# POT1 stability and binding measured by fluorescence thermal shift assays

**Lynn W. DeLeeuw**[1], **Robert C. Monsen**[1], **Vytautas Petrauskas**[2], **Robert D. Gray**[1], **Lina Baranauskiene**[2], **Daumantas Matulis**[2]*, **John O. Trent**[1]*, **Jonathan B. Chaires**[1]*

1 James Graham Brown Cancer Center, University of Louisville, Louisville, KY, United States of America, 2 Department of Biothermodynamics and Drug Design, Institute of Biotechnology, Life Sciences Center, Vilnius University, Vilnius, Lithuania

* j.chaires@louisville.edu (JBC); john.trent@louisville.edu (JOT); daumantas.matulis@bti.vu.lt (DM)

**Data Availability Statement:** All relevant data are within the manuscript and its Supporting Information files.

## Abstract

The protein POT1 (Protection of Telomeres 1) is an integral part of the shelterin complex that protects the ends of human chromosomes from degradation or end fusions. It is the only component of shelterin that binds single-stranded DNA. We describe here the application of two separate fluorescent thermal shift assays (FTSA) that provide quantitative biophysical characterization of POT1 stability and its interactions. The first assay uses Sypro Orange™ and monitors the thermal stability of POT1 and its binding under a variety of conditions. This assay is useful for the quality control of POT1 preparations, for biophysical characterization of its DNA binding and, potentially, as an efficient screening tool for binding of small molecule drug candidates. The second assay uses a FRET-labeled human telomeric G-quadruplex structure that reveals the effects of POT1 binding on thermal stability from the DNA frame of reference. These complementary assays provide efficient biophysical approaches for the quantitative characterization of multiple aspects of POT1 structure and function. The results from these assays provide thermodynamics details of POT1 folding, the sequence selectivity of its DNA binding and the thermodynamic profile for its binding to its preferred DNA binding sequence. Most significantly, results from these assays elucidate two mechanisms for the inhibition of POT1 –DNA interactions. The first is by competitive inhibition at the POT1 DNA binding site. The second is indirect and is by stabilization of G-quadruplex formation within the normal POT1 single-stranded DNA sequence to prevent POT1 binding.

## Introduction

The protein POT1 (Protection of Telomeres 1) [1] is an integral part of the shelterin complex that protects the ends of human chromosomes [2–4]. It is the only component that binds directly to the single-stranded DNA overhang. POT1 is anchored to the shelterin complex by its interaction with the protein TPP1 [5]. The TPP1-POT1 complex suppresses the ATR-dependent DNA damage response, and helps recruit telomerase to telomeres for DNA replication [6]. The role of the TPP1-POT1 heterodimer is complex and multifaceted [5–7]. The

**Funding:** Funded by grant GM077422 from the National Institutes of Health (JBC, JOT). https://www.nih.gov/grants-funding. The funders had no role in study design, data collection and analysis, decision to publish, or preparation of the manuscript.

**Competing interests:** The authors have declared that no competing interests exist.

heterodimer can inhibit telomerase binding as part of the shelterin complex, or can serve as a processivity factor for telomerase during telomere extension [8]. The single-stranded telomere overhang readily folds into G-quadruplex (G4) structures that inhibit telomerase binding [9–11]. POT1 unfolds telomeric G4 structures by conformational selection [12] to render the telomeric overhang accessible to telomerase [10, 12–17]. Mammalian cells have 50–100 POT1-TPP1 molecules per telomere that can fully coat the ssDNA overhang to form a compact globular structure [16]. A study that combined ensemble and single-molecule telomerase assays suggested a synergy between POT1-TPP1 binding and G4 folding in support of telomerase function [18]. A number of germline mutations in POT1 were identified in cancers and other telomere syndromes [4, 6, 7, 19].

Human POT1 may be a potential drug target due to its role in telomere maintenance and telomerase function. The Wuttke laboratory sought small molecule inhibitors of POT1 binding to DNA by using a custom isothermal, time-resolved, Förster resonance energy transfer (FRET) assay to monitor oligonucleotide binding [20]. Screening of a chemical library of 20,240 compounds yielded a single inhibitor, Congo Red (CR). CR binding to POT1 was validated by isothermal titration calorimetry (ITC). The closely related bis-azo dye Trypan blue was then found to also bind to POT1, but with slightly lower affinity compared to CR. Recent virtual screening for POT1 inhibitors yielded several potential hits [21]. However, none of these hits were validated by any experimental assay, raising concern about their practical utility as inhibitors of POT1. POT1 directed drug discovery efforts are limited by the lack of convenient screening and validation tools to evaluate its DNA or small-molecule binding. POT1 binding to DNA typically is studied by laborious electrophoretic mobility shift assays, specialized single-molecule methods, or, rarely, ITC. Functional POT1 studies rely on complex biochemical approaches, like telomerase extension assays, that are difficult to implement as even moderate throughput methods. Here we use variations of the fluorescence thermal shift assay (FTSA) for the efficient screening of POT1 binding interactions [22–24]. FTSA is also known as differential scanning fluorometry (DSF), as the ThermoFluor® assay or as thermal denaturation fluorescence.(TDF).

Thermal shift assays have a long history. Schellman [25] and Peller [26] first described theories for the coupling of small-molecule binding to changes in the thermal or chemical denaturation transition midpoint for proteins. Thermal denaturation methods have been widely used to study DNA-small molecules interactions since the early 1960s [27] and the increase of the duplex DNA melting transition temperatures ($T_m$) became an accepted criterion for small molecule binding. Crothers and McGhee each developed statistical mechanical models that described shifts of duplex DNA melting transition temperatures resulting from small molecule binding [28, 29]. These theories used an Ising model for the DNA helix-coil transition, incorporated the effects of neighbor exclusion on binding and considered the effects of preferential ligand binding to either the helix or coil form that would result in either increases or decreases in $T_m$, respectively. These theories predicted multiphasic DNA melting transition curves at less than saturating ligand concentrations, resulting from ligand redistribution over the course of the melting transition. These theories also were used for the analysis of ultratight ligand binding to DNA [30, 31], and provide a basis for the development of assays to study the sequence- and structural-selective binding of small molecules to DNA [32, 33]. The practical use of thermal shift data for quantitative studies of protein-ligand interactions was codified in a remarkable paper by Brandts and Lin [34] in 1990, who derived equations for a number of typical binding models. These equations could be applied to extract binding constants from differential scanning calorimetry experiments. These early thermal shift approaches for both DNA and protein binding were thermodynamically rigorous, but were laborious and time consuming because of the need to acquire multiple melting transition curves over a wide range of temperatures and ligand concentrations in serial experiments.

A revolutionary development for the thermal shift assay came in 2001 with the advent of a high-throughput miniaturized version that used plate reader technology featuring small sample volumes and parallel monitoring of samples [23]. In this assay an extrinsic fluorescent dye was used to monitor the exposure of hydrophobic surfaces upon protein denaturation. This technology, dubbed ThermoFluor®, allows for the high-throughput screening of ligand binding to a target protein for drug discovery, and also for biophysical measurements of stability as part of structural genomic efforts [35]. The method was validated by comparison to independent calorimetric methods [36, 37]. Several approaches for the analysis of thermal shift data according to differing assumed reaction models have been proposed [34, 36, 38–42]. The ease of measurement, the minimal sample consumption and the sound thermodynamic basis of the method have led to the wide use of high-throughput thermal shift assay to study protein-ligand interactions and protein stability.

FRET assays were developed to monitor thermal shifts resulting from small molecule binding to G-quadruplex (G4) and other nucleic acid structures [43–45]. Instead, using an extrinsic fluorescent probe, FRET requires attaching a suitable donor-acceptor pair to the ends (or other defined sites) of the nucleic acid of interest, a labeling step that is greatly facilitated by modern DNA synthesis technology. Most often this approach is used to screen for small molecules that are selective for binding to particular nucleic acid structures, such as G4 structures [46]. We are unaware of any wide-spread use of this approach to study protein-nucleic acid interactions. One of our aims here is to show the application of FTSA for such a purpose, using POT1 as an example [12].

Scheme 1 shows the set of coupled POT equilibria, and the possible use of FTSA to illuminate individual steps in this reaction scheme. The scheme incorporates the denaturation of

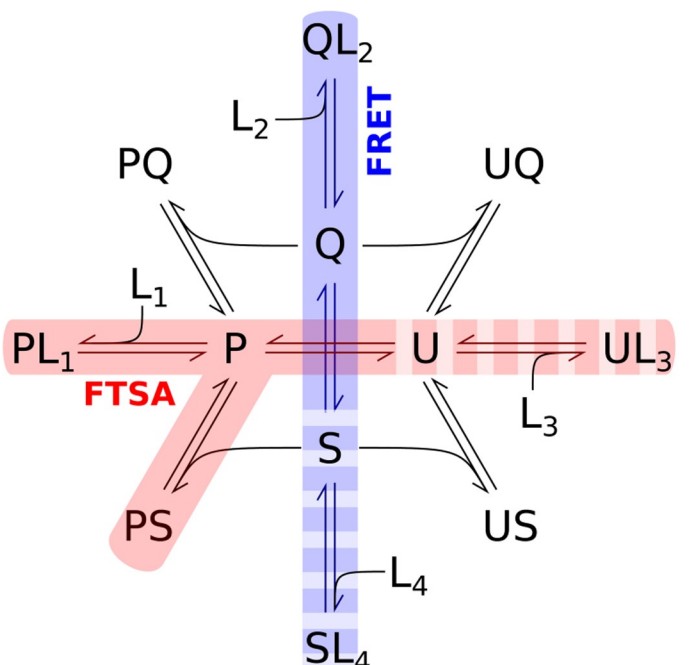

**Scheme 1. POT1 (P) binding equilibria and possible inhibition pathways.** U is unfolded POT1, Q represents folded G4 structures, S is unfolded (or single-stranded) G4 structures and L is a small molecule binding ligand. By using different detection probes, FTSA can monitor the fate of different reactants. Sypro Orange (SO) allows POT1 to be monitored, while the G4 is "invisible". Conversely, FRET-labeled G4 allows the G-quadruplex to be monitored, while POT1 is "invisible". Sypro Orange would be sensitive to POT1 denaturation, to stabilization by interactions with ssDNA and small molecule ligands FRET labeled G4 would directly sense G-quadruplex unfolding and stabilizing interactions arising from ligand binding or destabilizing interactions with POT1.

both the protein POT1 (P) and a G4 (Q), and considers possible small molecule ligand ($L_x$) binding to both the folded (P, Q) and denatured (U, S) macromolecular forms. The center of Scheme 1 shows the two-state denaturation events for POT1, $P \rightleftharpoons U$, and for the G4, $Q \rightleftharpoons S$. POT1 unfolding can be monitored by SyproOrange® fluorescence, while G4 denaturation can be monitored by FRET using a suitably labeled oligonucleotide. Different small molecule ligands might bind to the folded or unfolded forms of POT1 or G4, perturbing their thermal unfolding. The equilibria of primary interest to us are the interaction of POT1 with single-stranded DNA forms (S). The right side of Scheme 1 shows interactions of unfolded macromolecular forms with each other or with added small-molecule ligands. These interactions are shown for completeness but are not related to our studies. The left side of Scheme 1 shows the equilibria of POT1 binding to small molecule ligands or single-stranded DNA, with each reaction potentially detectable as thermal shift in protein stability. G4 stability, selectively monitored by FRET, can be altered by small-molecule binding or by interaction with POT1, with detectable thermal shifts.

We describe here FTSA assays that allow us to both detect and quantify specific steps in Scheme 1. These assays provide new tools for exploring POT1 binding interactions. Our goal is to use these tools to understand the thermodynamics of POT1 interactions and to provide mechanistic insights into the inhibition of POT1 binding by small molecules.

## Results and analysis

Fig 1 shows the thermal denaturation results of POT1 by (A) FTSA and (B) independent circular dichroism experiments. FTSA monitors the fluorescence of SYPRO™ Orange (SO) (which binds preferentially to denatured POT1 with increased fluorescence) as a function of temperature. Panels C and D in Fig 1 show the first derivative of these transition curves. The data show that POT1 melts with a sharp transition at $T_m$ = 51.5 ± 0.2°C (curve 1, panel C). Upon binding oligonucleotide O1, the known preferred POT1 binding sequence, the $T_m$ of POT1 is elevated to 64 ± 1°C (curve 2, panel C). A negative control oligonucleotide sequence does not bind to POT1 or appreciably alter the observed $T_m$ of the protein (curve 3, panel C). The insets in panels A and C show the near superposition of POT1 thermal denaturation in the presence of the negative control sequence. Panels B and D in Fig 1 show the thermal denaturation of POT1 monitored by circular dichroism in the presence (curve 2, panels B and D) or absence (curve 3, panels B and D) of oligonucleotide O1, yielding $T_m$ of 62°C and 50°C, respectively. The concentrations and molar ratios of POT1 and O1 differ slightly (by necessity) in this CD experiment from those used in FTSA. The excellent agreement of $T_m$ values by an independent biophysical approach validates the results obtained by FTSA.

FTSA allows us to examine the quality of individual POT1 preparations and its stability under different solution conditions. S1 Fig in (S1 File) shows FTSA data for 22 POT1 preparations. For POT1 alone, transition curves for most preparations show a $T_m$ of 51.5 ± 0.2°C, providing a measure of protein integrity (S1A Fig in S1 File). A few preparations (shown in red) show anomalous denaturation curves, indicating flawed preparations. FTSA does not provide the origin or an explanation of the anomalous curves, but the deviant behavior provides a quality control measure to avoid using those preparations for any further studies. S1B Fig in (S1 File) shows that FTSA also shows POT1 binding to its preferred oligonucleotide sequence. Most preparations show $T_m$ of 64 ± 1°C, but, again, some preparations show anomalous melting indicative of flawed preparations. These quality control measures provide validation of the consistency in POT1 preparations, for both protein integrity and functional activity.

S2 Fig in (S1 File) shows the second application of FTSA as a stability screen [22, 47] of POT1. A commercially available kit that provides an array of common cosolutes (each at a

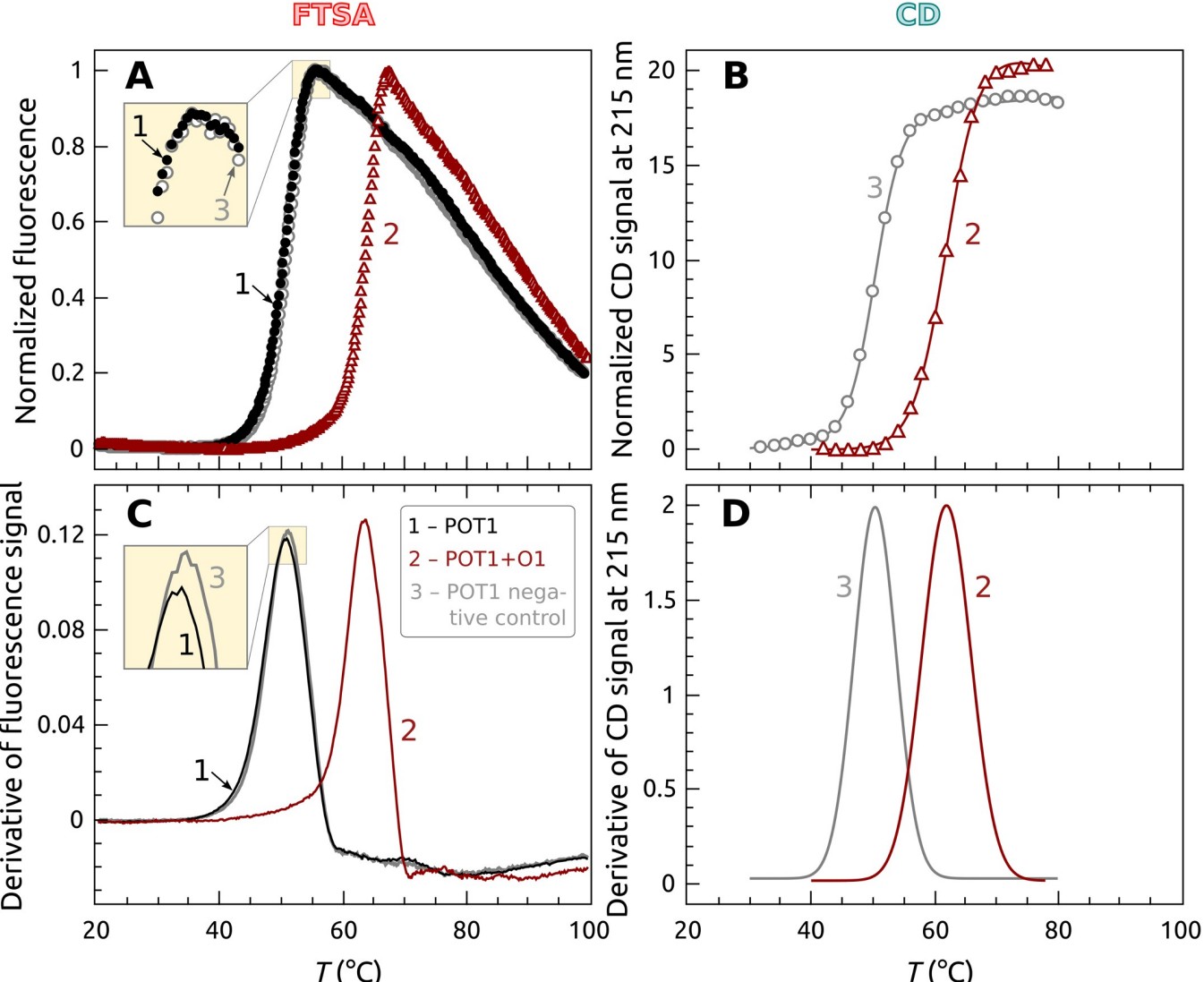

**Fig 1. Thermal denaturation of POT1 protein in the presence and absence of oligonucleotide O1 (5′ TTAGGGTTAG).** (A) Primary FTSA data using Sypro Orange. The black curve is POT1 alone, the red curve is POT1 with added O1 and the blue curve is POT1 with an added negative control oligonucleotide 5′ CTAACCCTAA. (B) Data obtained independently by circular dichroism to validate the FTSA. (C) The first derivatives (divided by $10^{-3}$) of the curves shown in panel A. (D) Derivative melting curves of the CD denaturation experiment from panel B. The black curve is POT1 alone, the red curve is for POT1 in the presence of excess O1. Conditions: [POT1] = 0.35 µM, [O1] = 0.5 µM.

single concentration) was used to measure the $T_m$ of POT1 under a wide variety of solution conditions. This provides a survey of solution conditions for biochemical and biophysical studies in which POT1 is stable.

Fig 2 shows FTSA results of POT1 titration with O1. The $T_m$ increases with increasing total O1 concentration. The titration curve is presented on a semilogarithmic scale to clearly show details over a wide concentration range. Features of the curve include first, little change in $T_m$ at substoichiometric concentrations of O1 ([O1]/ [POT1] < 0.1); second, a sharp increase in $T_m$ at O1 concentrations near the total POT1 concentration ([O1]/ [POT1] $\approx$ 0.2–1.0); and, finally, a gradual, nearly linear, increase in $T_m$ at O1 concentrations larger than the POT1 concentration ([O1]/ [POT1] > 1.0).

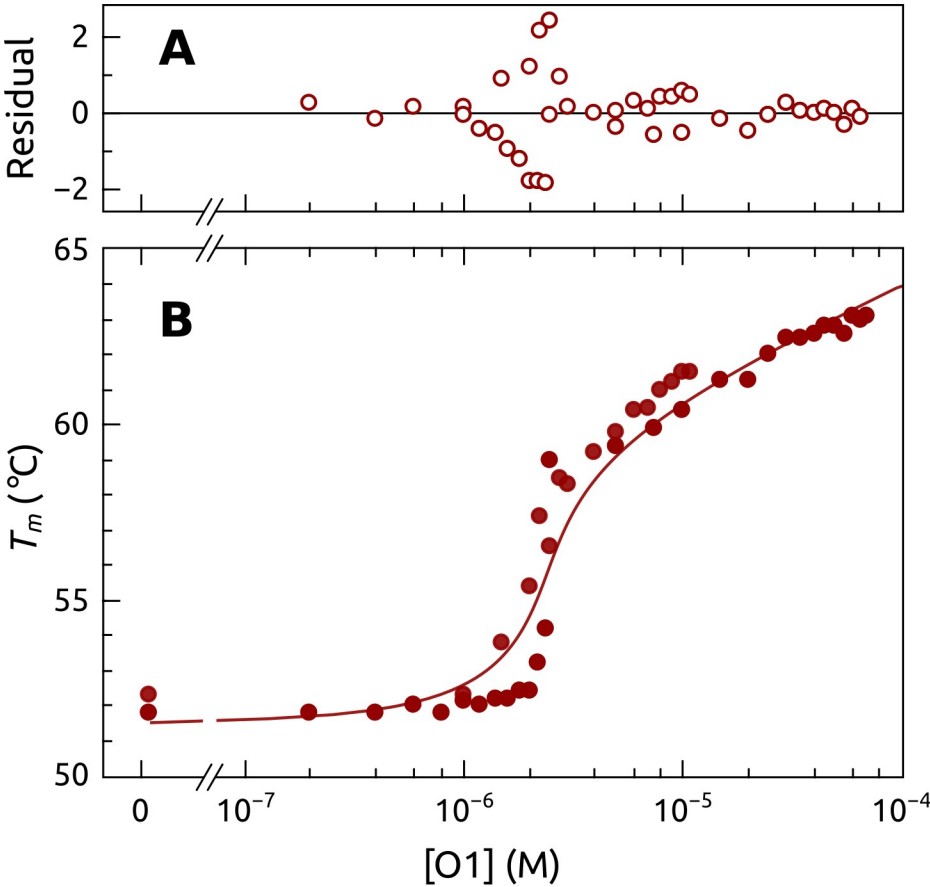

**Fig 2. Titration of POT1 with O1 measured by FTSA.** $T_m$ was measured as a function of added O1 concentration with constant [POT1] = 5 μM. The solid line shows the computed model using the parameters shown in Table 1. The residuals from the "best fit" model are shown in the top panel.

S1 Scheme in (S1 File) shows the simplest reaction model and equations to account for the behavior seen in Fig 2 [36]. Seven adjustable parameters are used to predict the change in $T_m$ as a function of ligand (O1) concentration, assuming that POT1 concentration is known. These parameters could be obtained by curve fitting methods, but in practice that approach is hampered by the high correlation between parameters and the dangers of overfitting. We modeled the data of Fig 2 by S1 Scheme in (S1 File) using an iterative approach in which some parameters were constrained to values that were obtained by independent measurements. For example, key parameters in S1 Scheme in (S1 File) are for the denaturation of the protein in the absence of ligand ($T_m$, $\Delta H^0_{U\_T_r}$, $\Delta S^0_{U\_T_r}$ and $\Delta C_{P\_U}$). S3 Fig in (S1 File) shows independent experiments used to estimate some of these parameters. Global fitting of FTSA POT1 denaturation curves in the absence O1 provided the most reliable estimates. In the global fit of 8 replicate experiments, we used an approach in which $T_m$ and $\Delta H^0_{U\_T_r}$ were shared for all data sets, but pre- and post-transition baseline parameters were optimized for each dataset. The results are shown in S1 Table in (S1 File), with $T_m$ = 324 ± 1K and $\Delta H^0_{U\_T_r}$ = 442 ± 2 kJ mol$^{-1}$, from which $\Delta S^0_{U\_T_r}$ = 1360 ± 7 J mol$^{-1}$ K$^{-1}$ was derived. The lower and upper 95% confidence limits of $\Delta H^0_{U\_T_r}$ are 438 and 446 kJ/mol, respectively. S1 Table in (S1 File) shows that fits to two denaturation experiments monitored by circular dichroism provide parameters in good agreement with those obtained by FTSA. The parameter $\Delta_u C_p$ was estimated by computation using

the known high-resolution structure of POT1 [48] and the ProtSA web application [49]. In this approach, the structure and sequence of the protein are used to generate an ensemble of unfolded structures, from which an ensemble of estimates for the change in polar and non-polar solvent accessible surface areas (SASA) upon denaturation is calculated. Empirical equations are then used to calculate $\Delta C_{P\_U}$. This method provided an estimate of $\Delta C_{P\_U}$ = 15.4 ± 0.4 kJ mol$^{-1}$ K$^{-1}$. The magnitude of this value is similar to $\Delta C_{P\_U}$ values measured by DSC (Differential Scanning Calorimetry) for other proteins with molecular weights similar POT1 [50]. Collectively, these approaches provide estimates of the key thermodynamic parameters for POT1 unfolding that can be constrained in quantitative models to interpret the data in Fig 2.

For O1 binding to POT1, a binding enthalpy $\Delta H^0_{b\_T_0}$ = -139 kJ mol$^{-1}$ was measured directly by ITC [12] and may be used as a constrained parameter. Table 1 shows available independent estimates of key thermodynamic parameters. Values for two adjustable parameters ($\Delta S^0_{b\_T_0}$, $\Delta C_{p\_b}$) in S1 Scheme in (S1 File) can be varied to model the FTSA data. Iterative refinement of these values yields the curve shown in Fig 2 with $\Delta S^0_{b\_T_0}$ = -245 J mol$^{-1}$ K$^{-1}$and $\Delta C_{p\_b}$ = -4300 J mol$^{-1}$ K$^{-1}$ (Table 1, Model 1). The residual plot in Fig 2 show that the model accurately describes the data. The value for the heat capacity change is at first glance surprisingly large, but is consistent with similar values found for other sequence-specific protein-DNA binding interactions [51, 52]. The pair of values chosen above do provide a minimum in the sum of squared deviations between the data and model, but other paired parameter values provide acceptable models for the data. For example, Table 1, Model 2 shows if it is fixed at $\Delta C_{p\_b}$ = 0 and the values of the binding enthalpy and entropy are iteratively refined, the match between the data and model are nearly as good as for Model 1. It is notable that the binding enthalpy obtained by this independent approach is in excellent agreement with the value obtained by ITC. Similarly, Model 3 (Table 1) shows that refinement of binding entropy and unfolding

**Table 1. Parameters used to model POT1-O1 FTSA titration curve in Fig 2.**

| Parameter[a] | Independently Measured | Model 1 | Model 2 | Model 3 | Units |
|---|---|---|---|---|---|
| $\Delta H^\circ_{U\_T_r}$ | 442 ± 2[b] | 442 | 442 | 442 ± 1 | kJ mol$^{-1}$ |
| $\Delta S^\circ_{U\_T_r}$ | 1360 ± 7[b] | 1360 | 1360 | 1360 | Jmol$^{-1}$ K$^{-1}$ |
| $\Delta C_{P\_U}$ | 15.4±0.4[c] | 15.4 | 15.4 | 15.4 | kJmol$^{-1}$ K$^{-1}$ |
| $T_m$ | 324 ± 1[b] | 324 | 324 | 324 | K |
| $\Delta H^\circ_{b\_T_0}$ | -139 ± 1[d] | -139 | -140 ± 30 | -139 | kJ mol$^{-1}$ |
| $S^\circ_{b\_T_0}$ | -325[d] | -245 ± 9 | -276 ± 111 | -275 ± 2 | Jmol$^{-1}$ K$^{-1}$ |
| $\Delta C_{P\_b}$ | not known | -4300 ± 1200 | 0 | 0 | Jmol$^{-1}$ K$^{-1}$ |
| $K_b$ | $3 \times 10^7$ [d] | $4 \times 10^{11}$ | $1 \times 10^{10}$ | $1 \times 10^{10}$ | M$^{-1}$ |
| $\Delta G^{25^\circ C}_b$ | -42 ± 1[d] | -66.1 | -57.7 | -57.9 | kJ mol$^{-1}$ |
| [POT1] | - | 5 | 5 | | µM |
| $\Sigma R^2/10^{-10}$ | - | 9.2 | 11.3 | 11.3 | Sum of squared residuals |

[a] Parameters are defined in S1 Scheme in (S1 File). Standard deviations of fitted parameters are shown. For the models, those parameters that were fixed are shown without error estimates, while the iteratively refined parameters show estimated standard deviations. For the models, a goodness-of-fit measure, the sum of the square of the residuals ($\Sigma R^2$), is shown, where R is the difference between the calculated model and the data points in Fig 2.

[b] S3 Fig and; S1 Table in (S1 File).

[c] Calculated using methods described in reference 49.

[d] Reference [12].

enthalpy provides another satisfactory match. In that case, the unfolding enthalpy agrees with estimates obtained by independent experiments.

Fig 3 shows the simulated results of $T_m$ as a function of added ligand concentration for various values of binding parameters that affect the magnitudes of $T_m$ shifts. These results emphasize that a given $T_m$ shift is not a direct measure of binding affinity, but is instead a more complex function of the enthalpy and entropy contributions to the Gibbs energy of binding. This leads to some nonintuitive precautions in the interpretation of thermal shift data. For these simulations, the values of the thermodynamic parameters for POT1 denaturation were fixed to their experimentally observed values (see caption to Fig 3), while binding parameters were systematically varied. Since the change in Gibbs energy $\Delta G_b = -RTlnK_b = \Delta Hb—T\Delta S_b$,

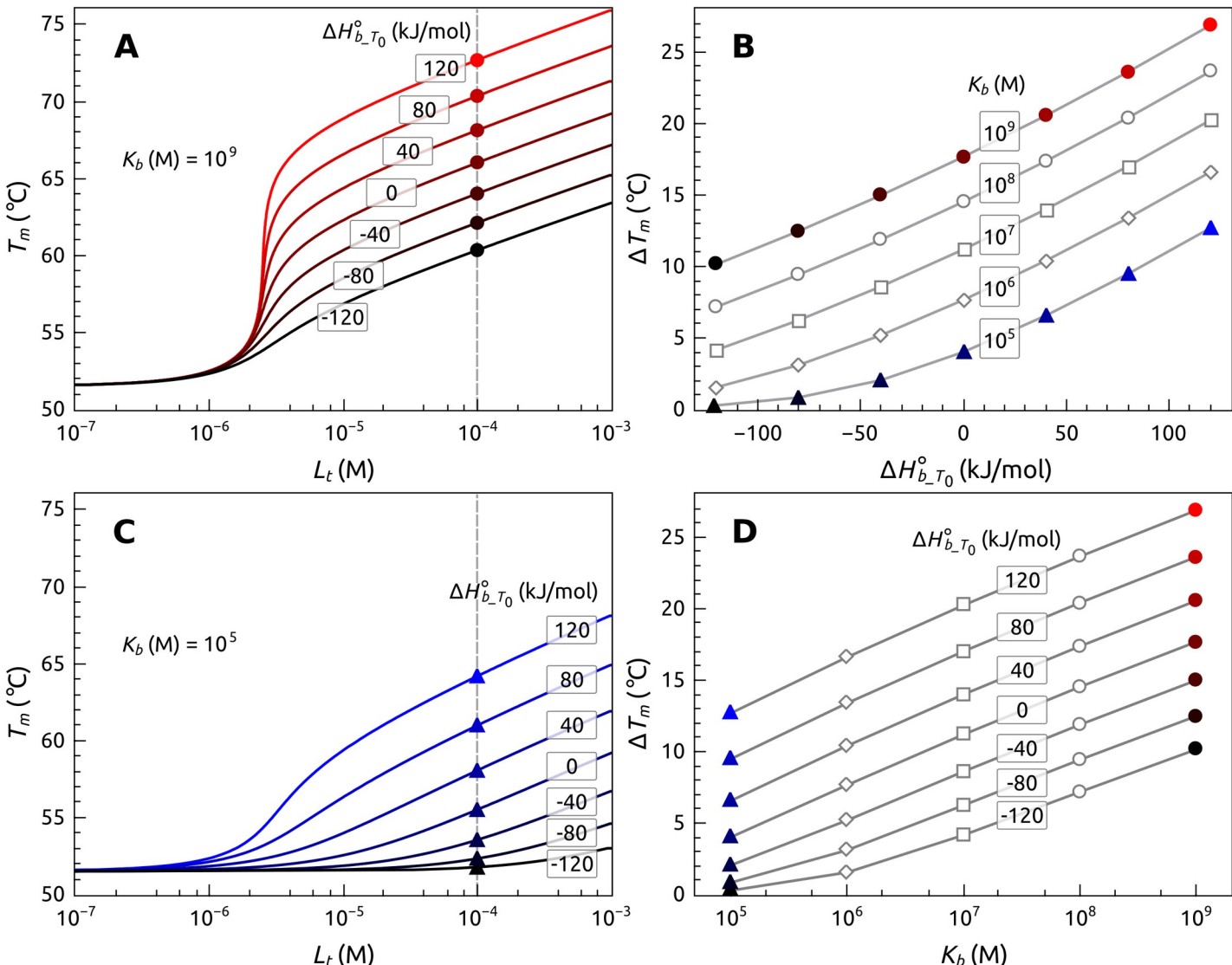

**Fig 3. Simulations of expected $T_m$ changes as a function of ligand binding enthalpy ($\Delta H_{b\_T_0}^0$) and association constant ($K_b$).** (A) Simulated titration curves for $K_b$ = $10^9$ M$^{-1}$ with enthalpy values varied from -120 to 120 kJ mol$^{-1}$. (C) Simulated titration curves for $K_b$ = $10^5$ M$^{-1}$ with enthalpy values varied from -120 to 120 kJ mol$^{-1}$. (B) $T_m$ shifts as a function of enthalpy ([POT1] = 5 μM; [Lt] = 50 μM). (C) The same $T_m$ shift can arise from many combinations of $K_b$ and $\Delta H_{b\_T_0}^0$. These simulations used: $\Delta H_{U\_T_r}^0$ = 44200 J/mol, $\Delta C_{P\_U}$ = 15000 J/mol-K, $\Delta S_{U\_T_r}^0$ = 1361.5 J/mol-K, $T_m$ = 51.5°C, $\Delta C_{P\_b}$ = -500 J/mol-K and [POT1] = 5 μM. Other parameters were varied as indicated.

there is an infinite number of enthalpy and entropy combinations that can lead to the same binding free energy, and therefore the same binding constant. The exact partitioning of binding enthalpy and entropy dictates the magnitude of the $T_m$ shift. Fig 3A shows FTSA titration curves for a constant binding free energy of 51.2 kJ mol$^{-1}$ ($K_b = 10^9$ M$^{-1}$) with different values for the binding enthalpy. Larger $T_m$ values are seen for positive binding enthalpies, for *entropically* driven binding reactions. The same effect is seen in Fig 3C for a lower ($10^5$ M$^{-1}$) binding constant. At a fixed ligand concentration, a large range in $T_m$ values may result for the *same* binding affinity, depending exactly on how the free energy is partitioned between the enthalpy and entropy terms. Fig 3B shows this nonintuitive behavior for a range of binding constant and enthalpy values, with $\Delta T_m$ values shown at a particular ligand concentration. If one traces a horizontal line at a fixed $\Delta T_m$ value, the same value can arise from different combinations of $K_b$ and $\Delta H^0_{b\_T_0}$. Thus, $\Delta T_m$ is *not* a direct measure of $K_b$. Fig 3D shows that $\Delta T_m$ values are correlated with $K_b$ *only if* the binding enthalpy $\Delta H^0_{b\_T_0}$ is constant. These simulations provide insight into the range of behavior expected for ligand binding to POT1 in FTSA experiments, and serve as a guide for the interpretation experimental $T_m$ changes as a function of ligand concentration.

An example of the utility of the FTSA for screening oligonucleotide binding to POT1 is shown in Fig 4. These data show the interaction of a series of oligonucleotide sequence variants with POT1. These sequences were taken from earlier studies from the de Lange laboratory [53] that used a far more laborious electrophoretic mobility shift assay (EMSA) to rank the affinity of these sequence variants in a semi-quantitative way. The $\Delta T_m$ values shown here were collected by FTSA, and show the effects of slight sequence variations, or of backbone type, on the oligonucleotide-POT1 interaction. These results are in agreement with the results of the earlier study [53], but were obtained faster, and with greater quantitative resolution than was possible with the EMSA approach. We show only the results obtained at a single oligonucleotide concentration here, as would typically be done for a rapid screening. More complete titration studies could be done to define the thermodynamic origins of the differences observed among sequence variants, but these are not within the scope of this study. The FTSA should be generally useful for exploring the sequence-selectivity of any protein-oligonucleotide interaction, and has so far been underutilized for that purpose.

Fig 5 shows results of a more complex POT1 interaction with G4. We reported a detailed study of that reaction, and showed that POT1 unfolds G4 structures formed by repeats of the human telomere sequence (TTAGGG)$_x$ by a conformational selection mechanism in which POT1 "traps" the single-strand form to shift the dynamic equilibrium to the unfolded, bound state [12]. Fig 5 shows that at equilibrium, the $T_m$ shift is similar to that observed for POT1 binding to its unstructured single-stranded preferred sequence. Recall that this assay focuses on the relative stability of POT1 alone or when complexed with DNA. These results show that regardless of the initial state, either a short oligonucleotide or a folded G4, the final POT1-DNA complex has a similar stability. Fig 5 shows, that not all G4 sequences and structures will react with POT1, only those with the requisite telomeric recognition sequence. The G4 structure formed by the modified c-myc promoter sequence (PDB: 1XAV) (Fig 5, curve 5) fails to bind to POT1, with little or no change in $T_m$. This assay was used in our previous study to show a broader range of G4 structures that failed to bind to POT1 [12].

Fig 6 shows the use of the FTSA to find inhibitors of G4-POT1 interactions, small molecules that might evolve into useful therapeutic agents. It has long been proposed that stabilization of G4 structures by small molecules might provide an avenue for drug development by blocking telomere maintenance by inhibition of critical protein-DNA interactions [54]. Direct demonstrations of that proposal are lacking. Fig 6 shows that addition of the porphyrin TMPyP4, a

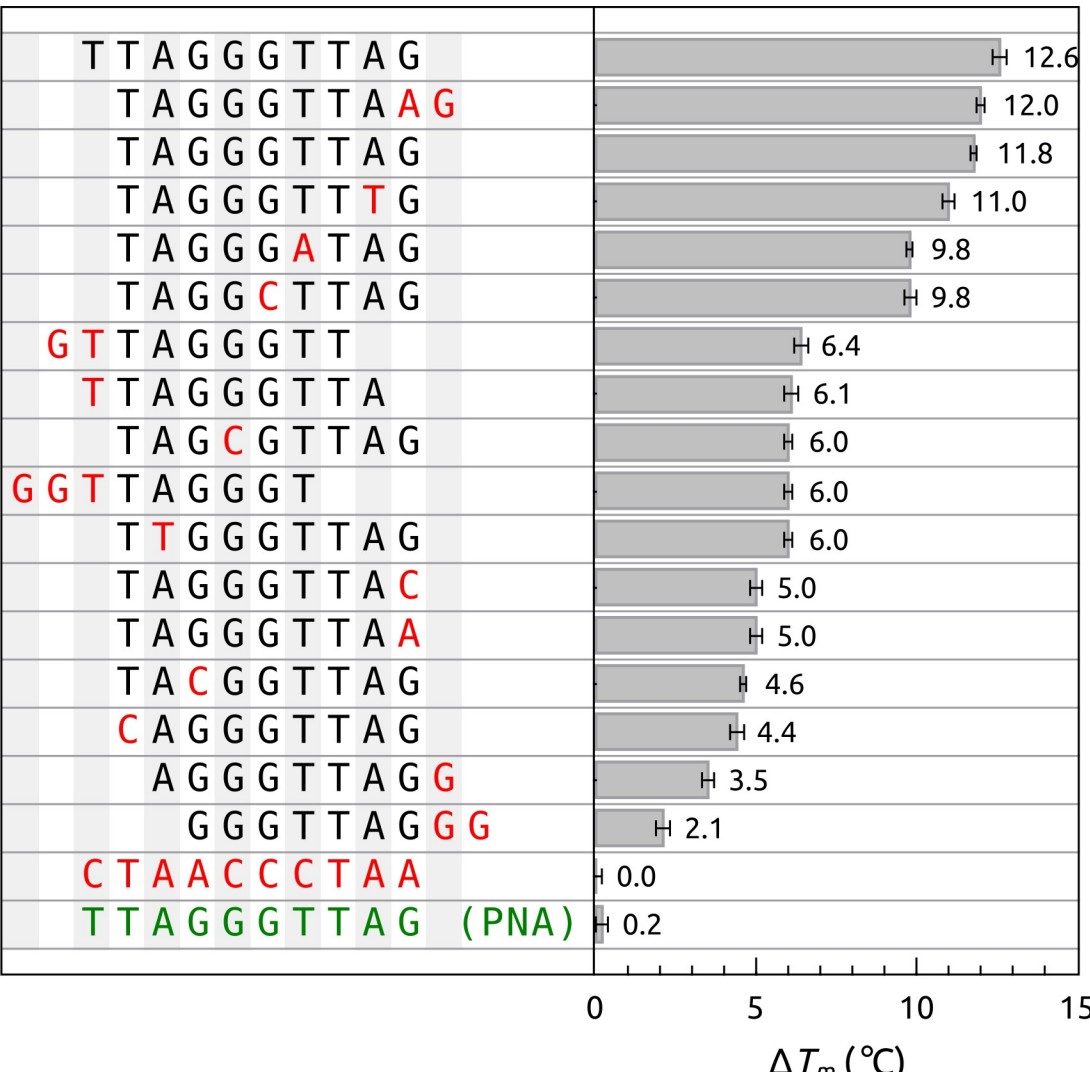

**Fig 4. Screening of the binding of oligonucleotide sequence variants to POT1 by FTSA.** The $\Delta T_m$ for POT1 binding to O1 (5′-TTAGGGTTAG) and several sequence variants is shown. Conditions: [POT1] = 5 µM and [DNA] = 50 µM. The PNA sequence shows a $\Delta T_m$ of only +1°C with [POT1] = 5 µM and [PNA] = 125 µM.

known G4 binder [55], inhibits G4 unfolding and consequently the formation of the POT1-DNA complex. Control experiments (not shown) indicated that TMPyP4 did not interact with POT1 directly, nor did it inhibit binding of the single-stranded O1 to POT1. The behavior seen in Fig 6 is best explained by a mechanism in which TMPyP4 binds to and stabilizes the G4 structure, preventing its unfolding to the single-stranded form that is required for binding by POT1. Such inhibition was also observed for the small molecules BRACO-19, Pyridostatin, and PhenDC3, all of which are known to bind to G4 [56].

Our implementation of the FTSA for POT1 was motivated, in part, for its use as an efficient screening tool for the discovery of small molecules that bound directly to POT1 to inhibit its DNA binding. Fig 7 provides a disappointing, but enlightening, cautionary tale about limitations of the FTSA. Congo Red (CR, S4 Fig in S1 File) is the only small molecule reported to bind to human POT1. The Wuttke laboratory [20], using ITC, reported that CR bound to POT1 with a 1:1 stoichiometry in an enthalpy-driven reaction with $K_a = 1.4 \times 10^6$ M$^{-1}$ and

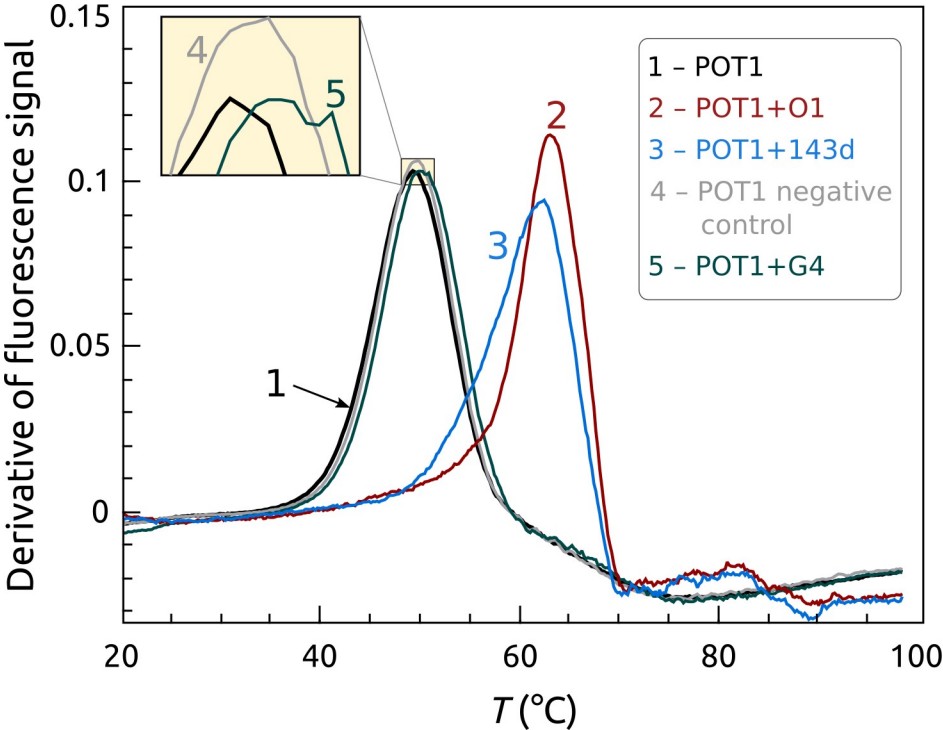

**Fig 5. Binding of POT1 to a human telomeric G4 structure.** POT1 binds to the initially folded telomeric sequence 5'AGGG(TTAGGG)₃ (143D) in a coupled reaction to form a single-stranded DNA-POT1 complex. The G4 is initially in a hybrid ("3+1") conformation under these conditions. FTSA monitors the $T_m$ shift resulting from the stabilization of POT1 upon complex formation. POT1 will not unfold, or bind to, a non-telomeric G4 structure. The curves are: POT1 alone (black), POT1+O1 (red), POT1+143D (blue), POT1 negative control (gray), POT1+G4 1XAV (green). POT1 = 5 μM and DNA = 50 μM.

$\Delta H_b$ = -10 kcal mol$^{-1}$ (41.8 kJ mol$^{-1}$). Fig 7 shows that addition of a 10-fold molar excess of CR to POT1 does not result in a discernable $T_m$ shift, from which it might be erroneously

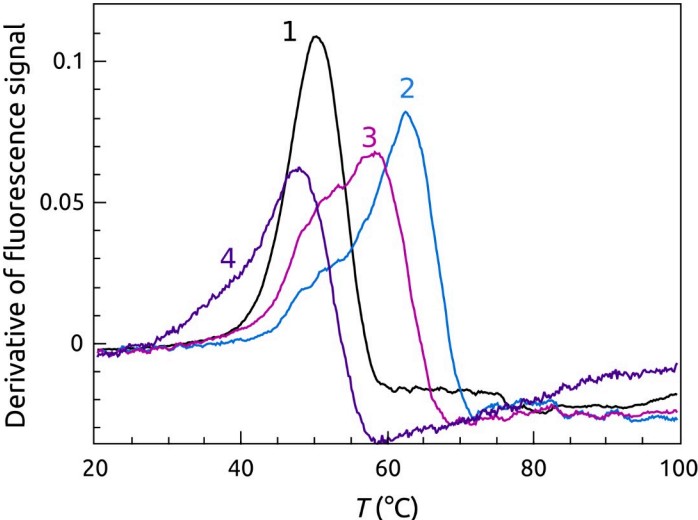

**Fig 6. Inhibition of POT1 binding to telomeric G4 structure by TmPyP4, a G4 binding small molecule.** The curves are: POT1 (1), POT1+143D (2), POT1+143D+ 50 μM TmPyP4 (3) and POT1+143D+ 250 μM TmPyP4 (4).

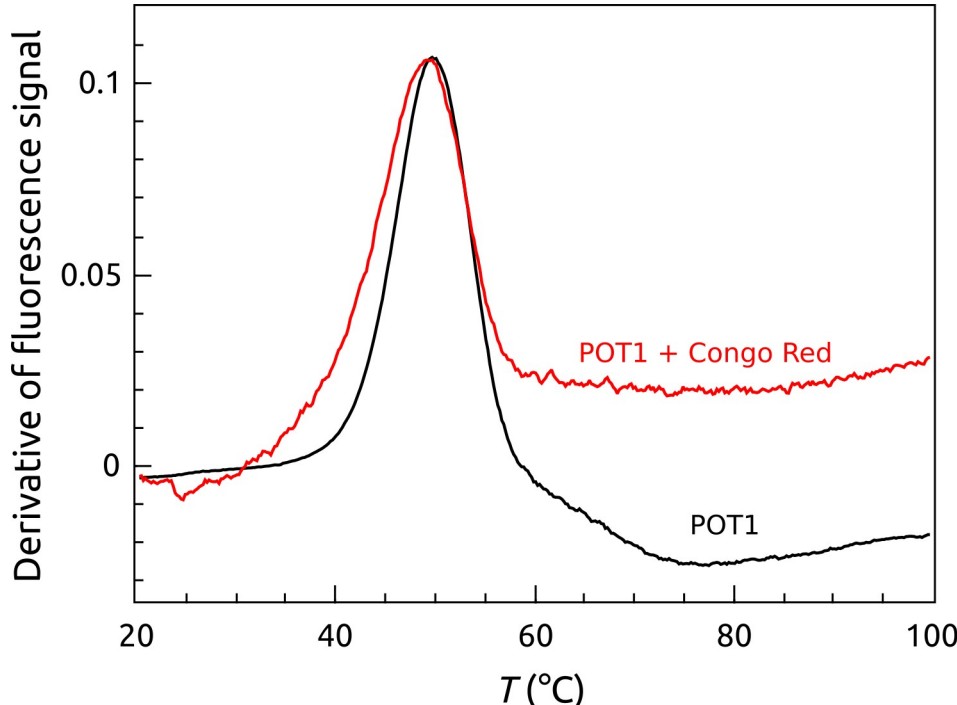

**Fig 7. Effect of Congo Red on the thermal denaturation of POT1.** There is no appreciable shift in the $T_m$ between POT1 (black curve) and POT1 + Congo Red (red curve). Conditions: [POT1] = 5 μM and [Congo Red] = 50 μM.

concluded that there is no binding. However, the simulations shown in Fig 3 taught us that moderately tight binders might show small $T_m$ shifts if the binding enthalpy is large and negative. Simulations using the reported CR thermodynamic binding parameters showed that under the conditions of the FTSA experiment shown in Fig 7, a $T_m$ shift of $\leq 2°$C would be expected, a change difficult to reliably discern. This finding emphasizes again that $T_m$ shifts are not direct measures of binding affinity, but are more complex metrics governed by the relative contribution of enthalpy and entropy to the binding free energy. We will show in a later section a different FTSA that does show the binding and inhibition of POT1 by CR.

We undertook an extensive virtual screening effort to find small molecules that bound to POT1 and inhibited its interactions with DNA. The FTSA was designed to validate virtual screening hits from the virtual screening effort. The results were disappointing, but are instructive and may inform future POT1 drug discovery efforts. Fig 8 shows the POT1 target used in screening [48]. Two docking sites were created within the Oligonucleotide Binding Domains (OBD) based on residues within the bound oligonucleotide ligand. The first docking site encompassed the 5'-d(TTAG), and the second was based on the d(GGG). Using the docking program Surflex-Dock we screened over 53 million virtual small molecules from the ZINC drug-like collection (2014, 2016, and 2018) at each of the two sites. The top 500 ranked molecules were pooled and clustered to remove redundant molecules and to identify unique scaffolds for use in screening. Initially, 72 compounds were purchased based on docking rank, chemical diversity, and availability. Each compound was tested by FTSA at multiple stochiometric ratios with POT1. In a second, separate virtual screening attempt, we utilized a combination of docking and MD simulation with post-hoc binding free energy calculations. To facilitate this process, we developed an automated script, Docking Free Energy Calculator (DFEC), which takes a ranked list of poses from a docking run and automatically performs

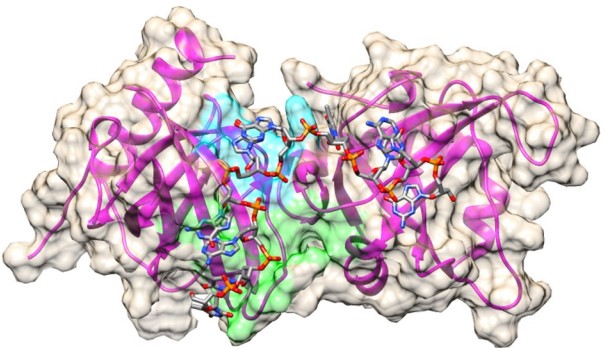

**Fig 8. Target sites on POT1 for virtual screening.** Initially two sites were chosen due to the size of the DNA binding area. POT1A site is shown in green on the surface and encompassed the oligonucleotide residues T1, T2, A3, and G4. POT1B site is shown in cyan on the surface and encompassed residue G5 from the crystal structure (PDB ID: 1XJV) deposited by the Cech laboratory [48]. Details of residue-based protomol generation in Surflex-Dock are provided in Methods.

explicit solvent MD simulations of receptor-ligand complexes and calculates the free energies of binding. The DFEC calculation was performed on 100 of the top-ranking virtual molecules from our original Surflex-Dock screen after re-docking each molecule into the two sites depicted in Fig 8. The calculated relative free energies ranged from >0 to -45 kcal/mol (1kcal = 4.184 kJ). The top 31 molecules below an arbitrary cutoff of -30 kcal/mol were individually assessed for their molecular structure and interactions with POT1 and 19 compounds were chosen for testing in the FTSA (we note that the scores from Surflex-Dock did not correlate with calculated relative free energies). From both virtual screening methods a combined total of 91 small molecules (72 + 19) were purchased and tested by FTSA. The distribution of $T_m$s for all 91 compounds are shown in S5 Fig in (S1 File).

The structures of representative hits are shown in S4 Fig in (S1 File). Unfortunately, none of these hits showed an appreciable $T_m$ shift by FTSA (within the limitation of detection shown in Fig 3), indicating an apparent lack of binding. For selected compounds, lack of binding was confirmed by more laborious biophysical tools, including ITC and analytical ultracentrifugation. While the failure to discover small molecule hits is disappointing, the FTSA worked as intended as a crucial validation step, and its use saved us from further efforts with compounds that did not hit their intended target with sufficient affinity.

We looked for other small molecules that might bind to POT1 by another approach. The crystal structure of POT1 in complex with O1 [48] indicates that its OBD domains interact directly with the sugar and base moieties of O1. Nucleotide analogues therefore might bind to the POT1 binding pocket. Using FTSA we found that the commercially available nucleotide analogues Decitabine, Gemcitabine, Fluorouracil, Cladribine, Clofarabine, Fludarabine, Capecitabine, Azacitidine, Pentostatin, and Nelarabine all increased POT1's $T_m$ by 1–1.8°C (S6 Fig in S1 File). These shifts are small, but significant, and indicate binding to POT1. Thermal shifts of 1–2 degrees could correspond to μM affinity depending on ligand concentration and ligand binding entropy. We conclude that nucleotide analogues bind to POT1, and may represent an avenue for further development.

POT1-DNA interactions were also explored by a second FTSA ("FRET-FTSA") in which a telomeric G4 was labeled with a FAM-TAMRA FRET pair at the 3' and 5' ends, respectively. This assay monitors the fate of the DNA during its interaction with POT1, and complements the SO assay described above that focuses on the protein. Fig 9 shows the results of the assay, which monitors the fluorescence intensity of the donor FAM as a function of temperature. In

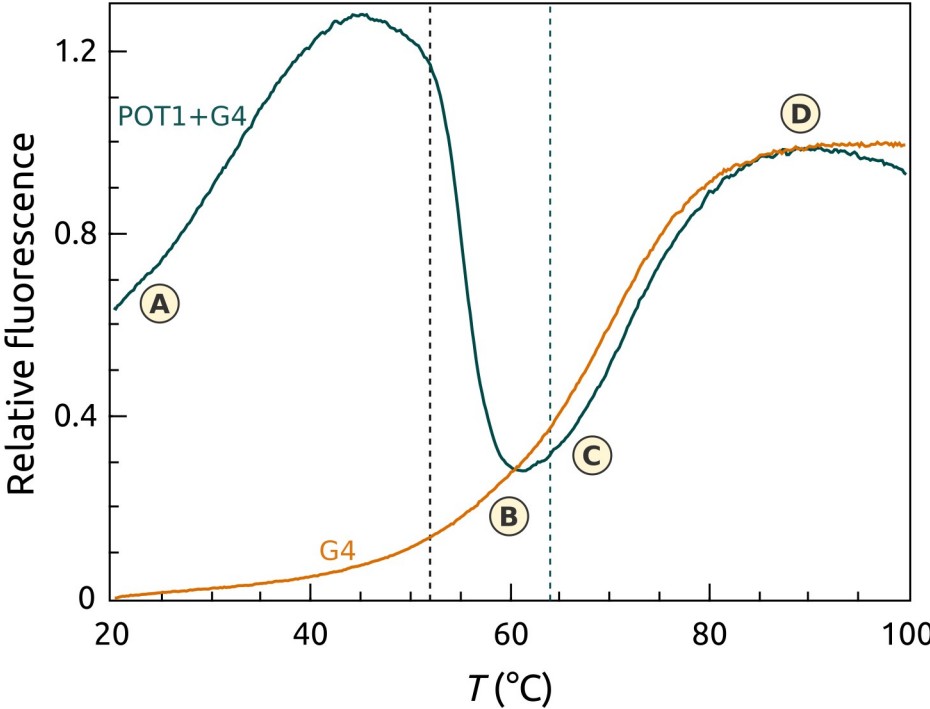

**Fig 9. FTSA using FRET-labeled G4 DNA in the presence (green) and absence (orange) of POT1.** This approach monitors the transition from the frame of reference of the G4 DNA. POT1 transitions would be invisible. The melting of the POT1-G4 complex is explained in the text. The dashed black vertical line shows the transition midpoint of POT1 from the protein frame of reference (Fig 1). The dashed green vertical line shows the transition midpoint of the POT1-G4 complex from the protein frame of reference (Fig 1).

the folded G4, the FRET pair is in proximity and FAM fluorescence is quenched. Upon denaturation, the ends separate and FAM fluorescence intensity increases. Fig 9 shows that in the absence of POT1, the denaturation of the G4 structures show a typical, sigmoidal shaped melting curve with a $T_m$ near 68˚C. The labels A, B and C in Fig 9 show sequential events in the POT1 interaction with initially folded telomere quadruplex. Point A shows the effect of added POT1 on the folded quadruplex at the start of the thermal denaturation. POT1 has unfolded the quadruplex, separating the distance between the donor-acceptor pair, resulting in an increase in the fluorescence signal. As temperature increases, fluorescence continues to rise until at just over 50 degrees it drops to point B near 61˚C. Recall from the Sypro Orange FTSA POT1 assay described above (Fig 1) that this is the temperature where the POT1-DNA complex denatures. Since the quadruplex is stable at this temperature (Fig 9), it refolds once POT1 is denatured, resulting in a decrease in the fluorescence signal as the FRET pair again comes into proximity. Point C then represents the unfolding of the refolded quadruplex at its normal melting temperature in presence of the denatured POT1. The pictographs at the top of Fig 9 show the progression from the denaturation of the POT1-DNA complex, the refolding of the DNA to the G4 form and the subsequent denaturation of the refolded G4.

Fig 10 shows FRET-FTSA that demonstrates three ways to inhibit POT1 unfolding of G4 structures. Fig 10A shows inhibition of POT1 by G4 stabilizing ligands, in this case BRACO-19 which binds tightly to G4 [56]. Addition of BRACO-19 stabilizes the G4 and elevates the $T_m$ by more than 20 degrees. The G4-BRACO-19 complex is recalcitrant to unfolding by POT1. In Fig 10B, G4 unfolding by POT1 is inhibited by addition of oligonucleotide O1, which itself binds to the POT1 DNA binding site and thereby acts as a competitive inhibitor at a 1:10

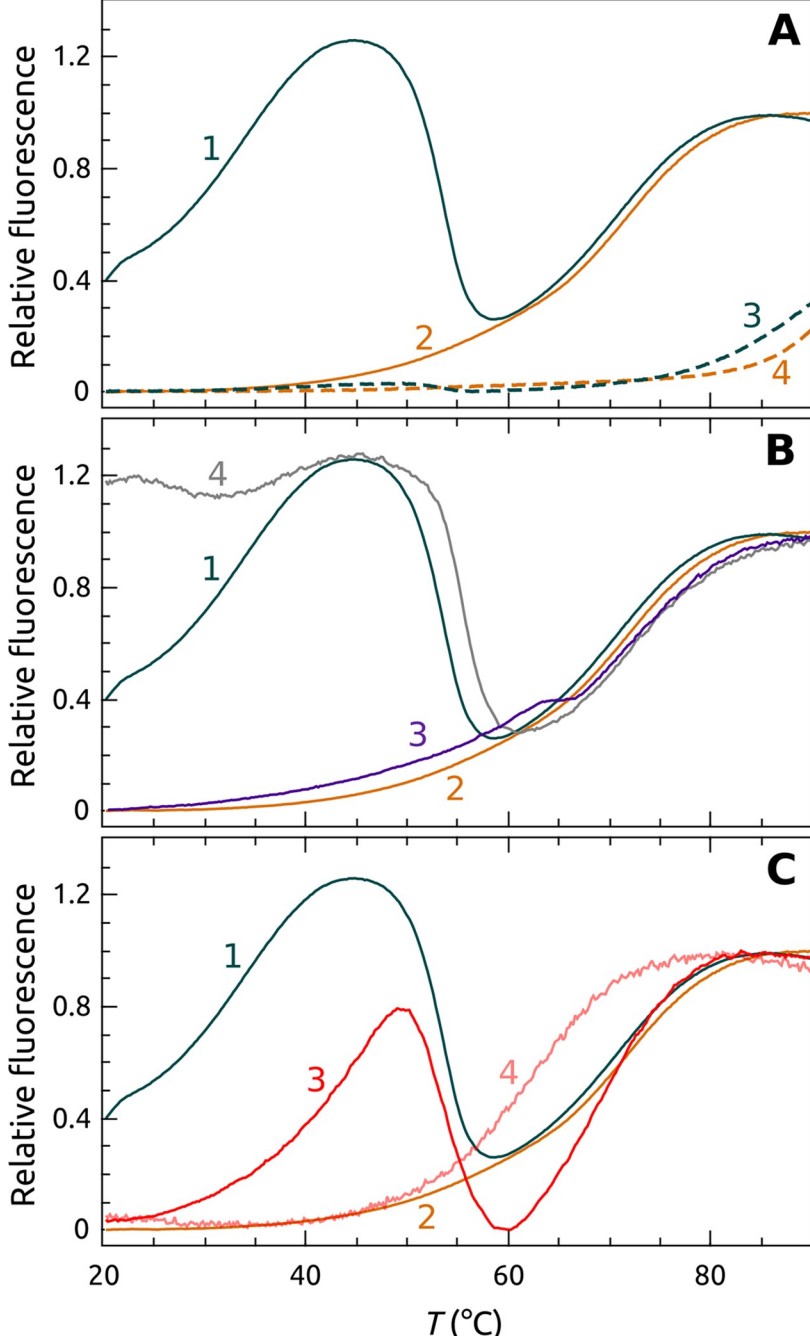

**Fig 10. Three ways to inhibit POT1-G4 unfolding.** (A) Inhibition of POT1 binding and unfolding by stabilizing G4 by ligand binding. The orange curve shows denaturation of G4 alone. The curve 1 shows the complicated melting of the POT1-G4 complex. Curve 2 shows melting of the G4 alone. Curves 3 and 4 show the denaturation of POT1 + G4 and G4 alone, respectively, in the presence of the G4 stabilizing ligand BRACO 19 at 1:100 (G4:BRACO 19). (B) Inhibition of POT1 by addition of 5′ TTAGGGTTAG (O1), which binds competitively to POT1 to prevent G4 binding and unfolding. The curves 1 and 2 are the same as panel A. The curve 3 shows the denaturation curve for G4 + POT1 +O1 at a ratio 1:10. The l curve 4 is a control using the oligonucleotide 5′ CTAACCCTAA which does not bind to POT1. (C) Inhibition of POT1 by Congo Red (CR). The curves 3 and 4 show mixtures of G4+POT1+Congo Red. at 1:100 (POT1:CR) and1:10 (POT1:CR), respectively. Curve 1 is the POT1 control; curve 2 is the G4 alone control.

(POT1:O1) molar ratio. The negative control oligonucleotide 5ʹ CTAACCCTAA used in the Sypro Orange FTSA shows some surprising behavior (Fig 10B, curve 4). Since its sequence is complementary to that of the G4, it is able to unfold the G4 as judged by the increase in fluorescence at 20°C, as we have previously described [12]. As the temperature increases, POT1, however, displaces the oligonucleotide to form the usual complex. Upon denaturation of the POT1-DNA complex, the G4 refolds, no doubt because the temperature is too high to permit stable binding of the short oligonucleotide. Lastly, Fig 10C shows that Congo Red can inhibit POT1 action. Fig 10C (curve 3) shows that at a 1:10 (POT1:CR) molar ratio, the unfolding of the G4 by POT1 is partially inhibited. At a 1:100 molar ratio (Fig 10C, curve 4), however, G4 unfolding is completely inhibited. There is a decrease in the apparent $T_m$ of the G4 in this ternary mixture. Explanation of that decrease would require further exploration but could plausibly reflect the complex equilibria that must exist in the ternary mixture, which must include temperature-driven dissociation of CR from POT1 (because its measured binding enthalpy is negative [20]).

## Discussion

While the structures of POT1 and telomeric G4 are known, it is essential to have companion kinetic and thermodynamic information to fully understand their stability and the mechanism of their interaction within the shelterin complex. This study focuses on the equilibrium thermodynamics of POT1-G4 interactions. Our results provide mechanistic insight into POT1 binding to DNA and demonstrate ways to inhibit that interaction.

The fluorescence thermal shift assays provide a versatile tool for biophysical studies of the POT1-DNA interaction. We show how FTSA can be tuned by the choice of fluorescent labels to illuminate specific aspects of the coupled equilibria that define POT1-DNA interactions (Scheme 1) to provide biophysical insights. FTSA provides new information about POT1 stability, the binding of DNA to POT1, and provides a tool for understanding the inhibition of the POT1-DNA interactions by small molecules. Each of these will be discussed in turn.

First, quantitative analysis of FTSA data requires an understanding of the fundamental thermodynamics of the folding of the "receptor" macromolecule, POT1 in our case. While this information is best obtained by DSC, that is not always feasible for proteins available in only limited amounts (like POT1). We show in S1 Fig and S1 Table in (S1 File) the thermodynamic parameters of POT1 unfolding as determined by FTSA and validated by independent CD measurements. The thermodynamic parameters for unfolding in S1 Table in (S1 File) were supplemented by an estimate of the heat capacity change for the unfolding of POT1 ($\approx$15000 Jmol⁻¹K⁻¹) computed from its structure. These thermodynamic parameters combine to provide an estimate for the stability of POT1 at 25°C of $\Delta G$ = -19.2 kJ mol⁻¹ for an assumed two-state unfolding reaction. No quantitative estimates of POT1 stability were previously available, nor was the thermodynamic profile for POT1 denaturation known. These thermodynamic values (S1 Table S1 File) are typical of a protein the size of POT1 [50]. This characterization is essential for the subsequent quantitative analysis of FTSA data, but is also of great practical utility on its own. These estimates for the enthalpy and heat capacity change for POT1 unfolding can be used to calculate a stability curve over a wide temperature range [57]. That curve provides the insight that POT1, in the absence of any stabilizing cosolutes, is subject to cold denaturation at a temperature midpoint of -3.4°C and is maximally stable at 23.4°C. This information is of practical use for the production and storage of POT1. We show in S1 and S2 Figs in (S1 File) how thermal denaturation can be used as an efficient quality control measure for POT1 preparations, and as a tool to optimize POT1 stability by the addition of cosolutes.

Second, the FTSA data shown in Fig 2 provides an independent biophysical characterization of POT1-DNA binding interactions. Compared to other biophysical approaches, this characterization required minimal sample consumption since it can be done at lower POT1 concentrations and in small reaction volumes. The information content of the data in Fig 2 is high, and yields a thermodynamic profile for the binding interaction (Table 1). The optimized thermodynamic model is consistent with estimates of the binding enthalpy ($\Delta H_b$ = -139 kJ/mol) obtained by ITC [12]. A novel insight provided by the analysis is that POT1 binding to DNA seems to be accompanied by a negative heat capacity change, $\Delta C_{p\_b} \approx$ -4000 Jmol$^{-1}$K$^{-1}$, a value similar to those found for other sequence-specific protein-DNA interactions [51, 52]. This detailed thermodynamic characterization provides a foundation for other practical applications of FTSA to studies of POT1-DNA interactions, as shown in Figs 4 and 5. The advantage of FTSA in this case is that the conformational selection mechanism that describes that reaction [12] can be viewed from the frame of reference of POT1 with the signal specific to the behavior of the protein within the more complex multistep equilibria. The $T_m$ shift observed is essentially the same as that seen for binding to the shorter single-stranded oligonucleotide O1, indicating that POT1 binding to the longer single-stranded G4 sequence, per se, has similar thermodynamics. The apparently lower overall affinity of POT1 for initially folded G4 structure is solely a consequence of the mandatory coupling of POT1 binding to an energetically unfavorable G4 unfolding step, as described in the conformation selection model [12].

Third, the FTSA assay can monitor the effects of small-molecule binding interactions on the formation of POT1-DNA complexes. G4 DNA has been proposed as a novel drug target [46, 54, 55]. The hypothesis proposed to find small-molecules that selectively stabilize G4 structures and prevent their unfolding, thereby inhibiting the formation of the single-stranded sequences required for protein recognition events which are important for gene regulation or telomere maintenance. There have been few demonstrations directly confirming that hypothesis, although there have been many indirect indications that it is correct by methods such as telomerase inhibition assays. Fig 6 shows a direct confirmation of the hypothesis. The porphyrin TmPyP4 is a known G4 binder, and its binding can stabilize the thermal denaturation of G4 by more than 20 degrees. Our results show that TmPyP4 prevents G4 unfolding and the formation of the POT1-DNA complex, as predicted.

Our quantitative analysis of FTSA data for POT1 interactions provides insights into the complexity and often nonintuitive behavior of thermal denaturation assays. Simulations of FTSA data, based on the thermodynamic profile of POT1 unfolding, guide subsequent interpretation of POT1 binding results for either DNA or small molecules. First, observed $T_m$ shifts are _not_ a direct measure of binding affinity, but are rather a function of the thermodynamic parameters of binding and thermal unfolding. In the G4 community, thermal denaturation methods are frequently used to identify and rank G4 binding ligands for drug discovery [58, 59]. It is important to recognize that differences in $T_m$ shifts reflect differences in the affinity of ligands only if they have identical binding enthalpies, as shown in Fig 3. In addition, $T_m$ shifts do not directly reflect the binding selectivity of a particular ligand for different G4 structural forms, since each G4 will have a unique unfolding thermodynamic profile that contributes to the magnitudes of thermal shifts. Second, these simulations show, that the magnitude of $T_m$ shifts is profoundly dependent on the binding enthalpy. Two ligands with identical binding constants might show dramatically different $T_m$ shifts depending on the exact partitioning of their binding free energies between the enthalpic and entropic components. Some analytical strategies attempt to extract "$K_d$" values for ligand binding from concentration-dependent $T_m$ values using simple bimolecular binding models that neglect to include enthalpy and entropy [60]. This strategy is suspect, and the "$K_d$" values obtained are subject to large systematic uncertainty. Indeed, if the data in Fig 2 are analyzed by such a model (S7 Fig in S1 File) an

association constant for POT1-O1 complex formation of $1.5\ (\pm0.5) \times 10^6$ M$^{-1}$ results. This is over three orders of magnitude lower than estimates obtained by other biophysical methods, or by FTSA. Another consequence of enthalpy neglect is a tendency, common in the G4 community, to erroneously assume that small $T_m$ shifts necessarily indicate low binding affinity. High affinity ligands with large, negative enthalpy values might show modest $T_m$ shifts. This mistake could lead to erroneous dismissal of promising ligands in drug discovery efforts.

FRET-FTSA reveals POT1 stability and binding from the DNA frame of reference for the more complicated case in which the DNA is initially folded into a G4 structure. This complementary assay has a somewhat more complicated readout, but illuminates multiple steps within the conformational selection mechanism [12] that governs this overall interaction. The assay serves as an isothermal indicator of POT1 unfolding of the G4 structure and complex formation, but then provides signals that follow the fates of both the POT1-DNA complex and the G4 DNA over the course of thermal denaturation of the complex and its components. This assay is particularly useful for visualizing different ways to inhibit POT1 unfolding of G4 DNA. First, the hypothesis that small-molecule stabilization of G4 can inhibit POT1 binding is validated. BRACO 19, a proven G4 binder [61], is seen to completely inhibit POT1 interactions and to dramatically elevate the $T_m$ of G4 denaturation. Second, the competitive inhibition of POT1 unfolding of G4 DNA by the oligonucleotide O1 was observed. Finally, FRET-FTSA shows Congo Red inhibition of POT1 binding, whereas the inhibition did not register (for the reasons discussed above) for the FTSA shown in Fig 7.

Our direct targeting of POT1 by virtual screening methods, although rigorous, was not successful, but the FTSA did work as intended to confirm (or not) hits. Our experience shows the critical importance of coupling virtual screening with an experimental validation. Our efforts also highlight the importance of the sampling protocol and the scoring algorithm in virtual screening. Even though Congo Red and the nucleotide analogues were present in the ZINC drug-like library they were not ranked in the initial top 500 and so were therefore not considered for testing. If more compounds were sampled these may have been identified. Testing of compounds in virtual screening is always a trade-off of price and availability of compounds. Secondly, the scoring algorithm did not place the nucleotide analogues highly in the rank order. In retrospect this is not surprising as most scoring algorithms are trained or tested on interactions found in protein-ligand complexes. If examples of small molecules binding to oligonucleotide binding sites were not represented in the scoring algorithm development it is unreasonable to expect the scoring algorithm to replicate binding affinity. The computationally intensive DFEC approach, which refined the scoring of only the top 100 compounds, is unable to predict tight binding compounds if they were not among the highest ranked compounds. Our learning experience suggests that in future attempts at virtual screening, several scoring approaches should be used, or that scoring approaches should be trained to reproduce small molecules binding to ssDNA binding sites on proteins, of which, unfortunately, there are few reported examples.

Is POT1 undruggable? To date, reports of successful virtual screening campaigns targeting ssDNA binding proteins in their oligonucleotide binding domains are sparse [62]. In fact, there is only one human protein structure, Human replication protein A, in the PDB of a single-stranded DNA OBD domain with ligands co-crystalized in the receptor site [63]. Recently, the Turchi lab, using an *in vitro* screen of more than 2,000 small molecules from ChemDiv and NCI libraries has identified ligands that bind at the OBD receptor sites with low µM affinities [64]. The only other OBD-ligand complex deposited is of a bacterial SSB, where the ligand is absent from the receptor OBD site [65]. No thermodynamic characterizations of ligand binding were reported from the RPA studies. Altogether, with the thermal shifts identified from our nucleotide analogues screens this would suggest that POT1 *is* druggable, but may require a

more nuanced approach if virtual screening is to be pursued, including different scoring approaches, nucleotide analogue enriched libraries, and consideration of receptor flexibility [62, 66].

## Material and methods

Oligonucleotides were purchased from Integrated DNA Technologies (Coralville, IA) on a 1-μmol scale with standard desalting. Lyophilized DNA was dissolved at a concentration of 0.5–1.0 mM in Milli-Q $H_2O$. Subsequent dilutions were prepared in 20 mM potassium phosphate, 180 mM potassium chloride, pH 7.2 to approximately 150–200 μM. Concentrations were determined by UV absorbance using the extinction coefficients supplied by the manufacturer. Quadruplex structures were annealed in a boiling water bath for 10 min then allowed to cool slowly overnight.

The FRET oligonucleotide 6-FAM-143D-TAMRA was obtained in HPLC-purified form from Sigma-Aldrich, St. Louis, MO. The FRET labeled quadruplex was dissolved in potassium buffer at a concentration of 100 μM. The labeled FRET quadruplex was further diluted to a concentration of 5 μM and annealed as above.

PNAs were purchased from PNA Bio, Inc. (Newbury Park, CA).

Compound Sources: TmPyP4, Congo Red, BRACO-19, Pyridostatin and Phen DC3 from Millipore Sigma and Sypro Orange from Thermofisher.

POT1 Compound Suppliers: Molport (Latvia), Ambinter (Orléans, France), Enamine (Monmouth Jct., NJ) and Mcule (Palo Alto, CA.). All compounds were dissolved to 10 mM in DMSO. Further dilutions were prepared in POT1 buffer.

### POT1 purification

The POT1-N coding sequence (hereafter referred to as POT1) was cloned into a pET21a expression vector designed to produce a protein with C-terminal 6xHis tag. After verifying the coding sequence, the protein was expressed in E. coli strain C41 as described previously [12]. For POT1 production, cells were suspended in M9 medium supplemented with 2% D-glucose, 10 ml/L of Basal Vitamins Eagle medium, 2 mM $MgSO_4$, 0.1 mM $CaCl_2$, and 100μM $FeSO_4$. Cultures were allowed to adapt to lower growth temperature (18˚C) for one hour with shaking in an incubator with a shaker. Expression of POT1 was induced by addition of IPTG to a final concentration 0.25 mM. After 16–18 hours cells were collected by centrifugation and either directly subjected to protein extraction or kept at -20˚C. The cell pellet obtained from 3 L of synthetic medium was suspended in 100 ml of lysis buffer containing 50 mM Tris-HCl, pH 7.2, 300 mM NaCl, 10% sucrose 10 mM imidazole, 0.1% NP-40, EDTA-free protease inhibitor cocktail, 1 mM PMSF, 2 mM β-mercaptoethanol. The suspension was sonicated once for 30 seconds (2 sec on/2 sec off) on ice to facilitate formation of homogeneous suspension. 2 mg of lysozyme was added directly to the suspension and incubated at 4˚C for 20 min on a nutator mixer. The extract was adjusted to 10 mM $MgCl_2$ and incubated at room temperature on a nutator mixer for 20 min after addition of 400 U of DNase I and 2 mg of RNase A, followed by sonication on ice for 30 sec (2 sec on/2 sec off) for a total of three times. We tested the efficiency of DNA digest/shearing by pipetting the extract up and down using yellow tip until no "strings" were detected. It is important to note that high viscosity of the extract will negatively affect following steps of purification. Cell debris was removed after centrifugation at 75,000 x g for 40 min and subsequently filtered through 20 μm filters. The POT1 was purified by immobilized metal affinity (IMAC) and anionic exchange chromatography. Briefly, we used an automated Profinia system (Bio-Rad) that allows sequential affinity chromatography and desalting of the sample with Bio-Rad cartridges, Profinity Ni-charged IMAC and P6, respectively.

Buffers, as 1X, were used in sequential order: A, 300 mM NaCl, 50 mM Tris-HCl, pH 7.2, 10 mM imidazole, 2 mM β-mercaptoethanol, 10% sucrose; B, the same as A, but 20 mM imidazole; C, the same as A, but 250 mM imidazole; D, (desalting buffer) 100 mM NaCl, 50 mM Tris-HCl, pH 7.2, 5 mM β-mercaptoethanol, 10% sucrose. The desalted sample was concentrated with Amicon centrifugal devices (10 kDa MW cutoff) and loaded onto a HiTrap Q HP anionic exchange column equilibrated in buffer D, using AKTA Purifier system. The vast majority of POT1 doesn't bind the resin and elutes in the flow-through fraction. The contaminating proteins bind the resin and can be eluted with increasing concentrations of NaCl. This procedure yields ~95% pure protein, as judged by SDS-PAGE after staining with Coomassie Blue. The purified protein was confirmed to be intact POT1-N by western blot analysis using 6xHIS-tag monoclonal antibody (H8), mass spectrometry, analytical ultracentrifugation (AUC), and by specific binding to the O1 oligonucleotide d[TAGGGTTAG] (S2 Table in S1 File). The purified protein was stored in aliquots of ~50 μM at -80˚C in buffer D. Prior to use, the thawed preparations were transferred into POT1 buffer (20 mM potassium phosphate, 180 mM KCl, pH 7.2) by gel filtration using a BioGel P6 spin column. POT1 concentrations were estimated from the absorbance at 280 nm using a calculated extinction coefficient of 41.37 mM$^{-1}$ cm$^{-1}$.

## Fluorescence thermal shift assay (FTSA)

For FTSA experiments, we used the Applied Biosystems StepOne Plus real-time PCR system. Melting curves were determined in 96 well plates using a melt curve temperature increment of 0.2˚C from 20˚C to 99˚C with the step and hold option (i.e. not continuous). Sypro Orange dye was used to label the POT1 protein at a final 1000-fold dilution (from 5,000x stock solution in DMSO). Ligand concentrations were tested at a minimum of 10-fold excess (50 μM) to the protein (5 μM). Later repeat screens included an increase of ligand concentrations from 100 to 250 μM, making proper adjustment of DMSO concentrations. The buffer solution was 20 mM potassium phosphate, 180 mM potassium chloride, pH 7.2. Twenty microliters of each sample were loaded in a 96 well plate, sealed and centrifuged at 1350 rpm for 2 minutes using an Eppendorf 5430 centrifuge equipped with a two-plate swinging-bucket microplate rotor. Each reaction was run in duplicate or triplicate and repeated on at least two different plates. POT1 alone and O1 DNA control were included in each plate to serve as a quality control check from batch to batch of protein. The $\Delta T_m$ was determined from the first derivative of the melt curve with and without ligand or DNA.

The same FTSA protocol was used for FRET labeled 143D DNA experiments. The concentration of FRET DNA was 0.5 μM, POT1 5 μM and ligand concentrations varied from 5 to 250 μM.

## Solubility and stability screen

The solubility and stability of POT1 was evaluated employing the Solubility and Stability Screen from Hampton Research, Aliso Viejo, CA. The protocol and reagents provided by the manufacturer were used to evaluate contributions of the different cosolutes. Briefly, 94 different reagents were provided in a 96 well plate. A protein stock solution was prepared in PBS at 0.15mg/ml POT1 and Sypro Orange dye at 1:1000. Ten μL of the plate reagents was then pipetted into a new 96 well plate with 10 μL of protein/dye solution. The analysis was conducted as described in the FTSA protocol above.

## Circular dichroism melting experiments

Temperature-dependent circular dichroism (CD) spectra of POT1 with and without O1 were measured in 1-cm or 3-mm path length quartz cuvettes using a Jasco J-810 CD spectrometer

equipped with a PFD-435S programable temperature controller. Spectral scanning parameters were 320–190 nm, 1.0 nm pitch, 1 nm spectral bandwidth, integration time of 4 s with three consecutive scans collected at each temperature. Temperature parameters were 20–80˚C, 2˚C pitch, 2˚C/min ramp, holding time of ± 0.1˚C for 5 s after reaching nominal temperature followed by a 60 s delay prior to wavelength scanning. A buffer blank consisting of 20 mM potassium phosphate, pH 7.2, 180 mM KCl was subtracted from the average of three consecutive spectra. $T_m$ and $\Delta H_{vH}$ values were estimated by non-linear least squares fitting the first derivative of the temperature dependence of the CD signal at 215 nm using the method of John and Weeks [67]. In addition, the temperature-wavelength data matrix in the wavelength range 280–205 nm was analyzed by singular value decomposition with fitting the amplitude vectors to a two-state unfolding model as described [68].

## Virtual screening

Virtual small molecule screening was performed using Surflex-Dock [69] with the KY Dataseam computing grid (http://www.kydataseam.com/). Initially the two Oligonucleotide Binding Domains (OBD) sites of POT1 from the crystal structure of POT1 OBDs bound to d (TTAGGGTTAG) were targeted (protein databank entry 1XJV, see Fig 8). The OBD1-T-TAGGG binding site was split into two docking sites based on the 5'-d(TTAG) and d(GGG) residues of the oligonucleotide ligand and protomols were generated. The first site was entirely in OBD1 and the second has partial overlap with OBD2 as OBD1 is the tighter and more selective DNA binding domain [48]. Small molecule libraries from the ZINC drug-like collection from 2014, 2016, and 2018 with 24,877,119, 17,244,856, and 11,154,739 compounds respectively, were docked at each site using our standard protocols [70]. This method has successfully found inhibitors to several proteins in the past. The top 500 compounds from each virtual screen were pooled and clustered based on Tanimoto similarity criteria to identify unique scaffolds using the Canvas application in Schrodinger's Maestro package [71, 72].

For the post-docking *in silico* relative free energy calculations we have created an automated script, Docking Free Energy Calculator (DFEC), which takes a ranked list of poses from a docking run and automatically performs free energy calculations for each compound. Here the top 100 ranked small molecules from the above Surflex-Docking were re-docked to POT1 into both protomol sites and subsequently passed to the DFEC script. Each complex was simulated for 3 nanoseconds of molecular dynamics using the AMBER's ff14SB [73] force field with TIP3P water parameterization with the radii set mbondi2. Ligand molecules were parameterized using the Antechamber [74] package with General AMBER Force Field (GAFF) [75] and AM1-BCC atomic charges [76]. MD simulations were equilibrated to 300 K at 1 atm of pressure using a standard procedure [12]. POT1-ligand simulations were analyzed using the CPPTRAJ module in the AmberTools18 package.

## Supporting information

**S1 File.**
(DOCX)

## Author Contributions

**Conceptualization:** Jonathan B. Chaires.

**Data curation:** Lynn W. DeLeeuw, Robert C. Monsen, Robert D. Gray, Jonathan B. Chaires.

**Formal analysis:** Robert C. Monsen, Vytautas Petrauskas, Robert D. Gray, Lina Baranauskiene, Daumantas Matulis, John O. Trent, Jonathan B. Chaires.

**Funding acquisition:** John O. Trent, Jonathan B. Chaires.

**Investigation:** Jonathan B. Chaires.

**Methodology:** Lynn W. DeLeeuw, Robert D. Gray, Jonathan B. Chaires.

**Project administration:** John O. Trent, Jonathan B. Chaires.

**Software:** Robert C. Monsen, Lina Baranauskiene, Daumantas Matulis, John O. Trent.

**Supervision:** Daumantas Matulis, John O. Trent, Jonathan B. Chaires.

**Validation:** Lina Baranauskiene, Jonathan B. Chaires.

**Visualization:** Robert C. Monsen, Vytautas Petrauskas, John O. Trent, Jonathan B. Chaires.

**Writing – original draft:** Jonathan B. Chaires.

**Writing – review & editing:** Lynn W. DeLeeuw, Robert C. Monsen, Vytautas Petrauskas, Robert D. Gray, Lina Baranauskiene, Daumantas Matulis, John O. Trent.

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
