## [Decision Letter · Decision Letter 0]

5 Mar 2021

POT1 Stability and Binding Measured by Fluorescence Thermal Shift Assays

PONE-D-21-00706

Dear Dr. Chaires,

We’re pleased to inform you that your manuscript has been judged scientifically suitable for publication and will be formally accepted for publication once it meets all outstanding technical requirements.

Kind regards,

Eugene A. Permyakov, Ph.D., Dr.Sci.

Academic Editor

PLOS ONE

Journal Requirements:

1. We note that you have included the phrase “data not shown” in your manuscript. Unfortunately, this does not meet our data sharing requirements. PLOS does not permit references to inaccessible data. We require that authors provide all relevant data within the paper, Supporting Information files, or in an acceptable, public repository. Please add a citation to support this phrase or upload the data that corresponds with these findings to a stable repository (such as Figshare or Dryad) and provide and URLs, DOIs, or accession numbers that may be used to access these data. Or, if the data are not a core part of the research being presented in your study, we ask that you remove the phrase that refers to these data.

Reviewers' comments:

Reviewer's Responses to Questions

**Comments to the Author**

1. Is the manuscript technically sound, and do the data support the conclusions?

Reviewer #1: Yes

2. Has the statistical analysis been performed appropriately and rigorously? 

Reviewer #1: Yes

3. Have the authors made all data underlying the findings in their manuscript fully available?

Reviewer #1: Yes

4. Is the manuscript presented in an intelligible fashion and written in standard English?

Reviewer #1: Yes

5. Review Comments to the Author

Reviewer #1: The Protection of Telomeres 1 (POT1) protein is an essential subunit of the shelterin telomere binding complex. It directly binds to single-stranded telomeric DNA, protecting chromosomal ends from an inappropriate DNA damage response. In this work thermodynamics of POT1 - DNA interaction is studied and two mechanisms of its inhibition are elucidated. In their work the authors used fluorescent thermal shift assay (FTSA) known also as differential scanning fluorometry. Excellent knowledge of the method and its modifications allowed the authors to obtain interesting new results on the thermodynamics of the POT1 interaction with DNA. I believe that the work is interesting not only due to the results obtained for POT1, but also as an example of the effective use of (FTSA). The work is well designed, performed and written and can be published as it is.

6. PLOS authors have the option to publish the peer review history of their article (what does this mean?). If published, this will include your full peer review and any attached files.

Reviewer #1: No

---

## [Editor Report · Acceptance letter]

12 Mar 2021

PONE-D-21-00706 

POT1 Stability and Binding Measured by Fluorescence Thermal Shift Assays 

Dear Dr. Chaires:

I'm pleased to inform you that your manuscript has been deemed suitable for publication in PLOS ONE. Congratulations! Your manuscript is now with our production department. 

Kind regards, 

on behalf of

Prof. Eugene A. Permyakov 

Academic Editor

PLOS ONE